# Transplant Candidates of 70+ Years Have Superior Survival If Receiving Pre-Emptively a Living Donor Kidney

**DOI:** 10.3390/jcm13071853

**Published:** 2024-03-23

**Authors:** Michiel G. H. Betjes, Marcia M. L. Kho, Joke Roodnat, Annelies E. de Weerd

**Affiliations:** Rotterdam Transplantation Institute, Department of Nephrology & Transplantation, Erasmus Medical Center, 3015GD Rotterdam, The Netherlands; m.kho@erasmusmc.nl (M.M.L.K.); j.roodnat@erasmusmc.nl (J.R.); a.deweerd@erasmusmc.nl (A.E.d.W.)

**Keywords:** elderly, kidney pre-emptive, transplantation, survival, mortality, dialysis

## Abstract

**Background**: The number of kidney transplant recipients over 70 years of age is increasing but detailed data on patient and graft survival in the modern era of immune suppression are few. **Methods**: A single-center cohort of patients of 70 years and older (n = 349) at time of kidney transplantation from 2010–2020 were followed until January 2023. **Results**: The median age was 73 years with a median follow-up of 4.3 years. Fifty percent of recipients of a living donor kidney (LDK, n = 143) received their graft pre-emptively. Cumulative death-censored graft survival was excellent in the LDK group and reached 98% at 5 years vs. 85% in the deceased donor kidney (DDK) group. Primary non-function (38%) and rejection (43%) were the major causes of graft loss in the first year after DDK transplantation. Rejection-related graft loss was 4.6% during follow-up. Median recipient survival was superior in the subgroup of pre-emptively transplanted LDK patients compared to non-pre-emptively LDK transplanted patients (11.1 versus 6.2 years). Non-pre-emptively transplanted patients had a significantly increased incidence of infection (HR 3.81, 1.46–9.96) and cardiovascular-related causes of death (HR 3.35, 1.16–9.71). Pre-emptive transplantation was also associated with a significantly improved graft survival in the DDK recipients but this result was confounded by significantly better HLA matching and younger donor age in this group. **Conclusions**: Pre-emptive LDK transplantation in patients of 70 years or older confers superior graft and recipient survival.

## 1. Introduction

Over the last decades significant progress has been made with regard to allograft survival in the first years after a kidney transplant by optimization of immunosuppression. In parallel, the number of kidney transplants performed in elderly ESRD patients has increased due to improved life expectancy. The proportion of transplant candidates aged 65 years and older continues to rise [1], and in the Netherlands, for example, the number of kidney transplant recipients of 65 years and above increased between 2006 to 2021 from 1181 (18% of the total number) to 4384 (36% of the total number), and for recipients of 75 years and above an even more striking rise from 163 to 1319 was noted (source: www.nefrovisie.nl/nefrodata (accessed on 30 November 2023)).

Analyses of large databases have consistently shown a survival advantage of kidney transplantation over remaining on dialysis in all age categories, also after correction of immortal time and lead time bias in more recent publications [2,3,4]. However, the number of recipients of 70 years and older is usually relatively small. Furthermore, most publications are derived from US databases [5] and there is a lack of detail on cause of transplant failure and transplant outcomes, which may differ substantially between Europe and the US [6].

After kidney transplantation, elderly recipients offer a number of medical challenges as they are more prone to side effects of immunosuppressive medication (e.g., post-transplant diabetes mellitus), have a higher infection rate and more frequently have a history of cardiovascular disease. This has a significant impact on recipient survival (apart from age itself) which is about 50% at 5 years, with death by infection and cardiovascular disease as major causes [7]. The allocation in Eurotransplant offers kidneys from donors of 65 years and above to 65+ recipients from the same region and often without HLA matching to allow for shortening of cold ischemia time (Eurotransplant senior program or “old for old program”). This has stretched the limits of kidney transplantation, resulting in about 10% non-functioning kidneys at 1 year post-transplantation and an eGFR below 30 mL/min in 30% of recipients [8]. However, survival was at least similar to the waitlist group, physical and mental health component scores improved significantly and the vast majority of recipients answered “yes” to the question of if they would do it again [9].

To further improve the outcome of the transplantation program for elderly patients it is necessary to have detailed data on recipients and kidney transplant outcome with regard to type of kidney transplant. In this study, a relatively large cohort of recipients of 70 years or older with half of them receiving a kidney from a living donor was analyzed for patient and kidney survival with data on cause of death and graft loss.

## 2. Materials and Methods

In this study, all recipients of a kidney transplant in the period January 2010 and December 2020 at the Transplantation Department of the Erasmus Medical Center in the Netherlands were included. On 1 January 2023 the database was closed for subsequent analysis. All recipients of a kidney transplant were seen at a regular basis at the out-patient clinic and their clinical data were recorded in the Dutch national database for organ transplants (Netherlands Organ Transplant Registry (NOTR)).

Kidney transplants were performed only in cases of a negative complement-dependent cytotoxicity cross-match in which both the current and historic sera were assayed. In cases of ABO blood group incompatibility a standard immune absorption procedure with induction therapy with basiliximab (Novartis Europharm, Basel, Switzerland) was given (and from 2015 onwards alemtuzumab (Genzyme Europe, Amsterdam, The Netherlands) [10]. The immune-suppressive medication consisted of induction with basiliximab. Tacrolimus (Astellas Pharma Europe, Leiden, The Netherlands) was started aiming for levels of 10–15 ng/mL in the first 2 weeks, thereafter lowering to 8–12 ng/mL and finally 5–8 ng/mL after 1 month. Tacrolimus was given in combination with mycophenolate mofetil (Roche Pharmaceutical, Basel, Switzerland) at a starting dose of 1 g b.i.d. and aiming for levels of 1.5–3.0 mg/L. Fifty milligrams of prednisolone was given b.i.d. intravenously on days 0–3. Thereafter, prednisolone was started orally at 20 mg which was tapered to 5 mg. At month 4–6, the use of prednisolone was discontinued.

The clinical and research activities being reported are consistent with the Principles of the Declaration of Istanbul as outlined in the “Declaration of Istanbul on Organ Trafficking and Transplant Tourism” and in accordance with the Declaration of Helsinki. All patients gave written informed consent for participating in the NOTR database.

No renal biopsies were performed as *per protocol* but instead the renal biopsies were *for cause*, namely when otherwise unexplained progressive loss of graft function or proteinuria occurred. The criteria of the 2018 Banff reference guide [11] were used to classify the renal biopsies. Independent of the presence of DSA or positive C4d staining, the biopsies that met the histological criteria for ABMR were classified as ABMR as has been carried out in previous studies [12,13] and has been discussed in detail before [12]. If a diagnosis of ABMR was made then pulse methylprednisolone with addition of intravenous immunoglobulins (1–2 g/kg bodyweight) was given. Only in cases of acute ABMR was plasmapheresis added to the treatment.

### 2.1. Outcomes and Variables

For data analysis the outcome of the kidney biopsy was further categorized as previously published [7] into five categories: TCMR, ABMR, recurrence of original kidney disease, diagnosis of de novo kidney disease and interstitial fibrosis with tubular atrophy (IFTA). In case of graft failure the diagnosis of the *for cause* kidney biopsy was used to categorize the type of graft failure if no other clinical event could explain the loss of kidney function. For example, if a patient with chronic ABMR with gradual loss of graft function had sepsis-related irreversible loss of graft function, then this case would be recorded as sepsis-related and not ABMR-related graft loss. The other primary categories of graft loss were a clinical diagnosis of cause for graft failure (e.g., heart failure, acute kidney injury etc.) and “unknown” if no biopsy was performed and no clinical diagnosis for allograft failure could be made (see Appendix A). Primary non-function is the category of grafts that never had function after transplantation because of acute tubular necrosis (ATN) which was diagnosed by (repeated) kidney biopsy.

### 2.2. Statistical Analysis

Differences in patient, donor and transplant characteristics were assessed by the Fisher’s exact test for categorical variables and Mann–Whitney U test for continuous variables. All *p*-values were 2-tailed. Death-censored graft loss and incidence of graft loss according to cause were assessed by Kaplan–Meier survival analysis with log-rank statistics for difference between strata. Univariate Cox proportional hazards analysis was used to identify relevant clinical and demographic variables as given in Table 1 for association with rejection and graft survival. Variables with a *p*-value of <0.1 were considered for stepwise forward regression to calculate hazard ratios and corresponding confidence intervals. Statistical analysis was performed with software IBM SPSS statistics 28.0.1.0, IBM corporation, New York, NY, USA.

## 3. Results

### 3.1. Baseline Characteristics and Recipient and Graft Survival

In total, 349 patients received a kidney transplant with a median age 73 years of which 143 (41%) received a living donor kidney and 103 (30%) patients were pre-emptively transplanted (Table 1). The vast majority of recipients (96%) were treated with basiliximab followed by triple immune suppression with tacrolimus as the calcineurin inhibitor of choice. Within a median follow-up period of 4.3 (2.6–6.4) years, 40% of recipients died with a functioning graft (Table 2). As expected, malignancies, infection and cardiovascular disease constituted the three main causes of death with a dominance of infectious-disease-related cause of death (27% of all causes) (Appendix A). Within the first year, 42% died of infections and 37% of cardiovascular disease.

**Table 1 jcm-13-01853-t001:** Clinical and demographic characteristics of 349 kidney transplant recipients of 70 years or older.

Median age recipient in years (IQR)	73 (71–75)
Male/female recipients	72%/28%
Median age donor in years (IQR)	65 (51–71)
Underlying kidney disease:	
- hypertensive nephropathy	44.4%
- glomerulonephritis	9.7%
- diabetes mellitus nephropathy	21.8%
- cystic nephropathy	4.8%
- other	12.3%
- unknown	7.0%
BMI (median, IQR)	27.0 (24–30)
Deceased/living donor kidney	59.1%/40.9%
- DBD type *	23.5%
- DCD type *	35.5%
- living related	14.3%
- living non-related	26.6%
Delayed graft function	25.8%
Pre-emptive transplantation	29.5%
Time on dialysis in years (median, IQR)	1.8 (1.2–2.6)
Cold ischemia time in hours (median, IQR)	6.5 (4.5–8.4)
Re-transplantation	8.8%
PRA > 5%	7.4%
HLA mismatches * (median)	3
Induction therapy	
Anti-IL-2 receptor antibody	96.4%
T-cell-depleting antibody	2.9%
Rituximab	0.7%
Maintenance immune suppression *	
- Steroids/tacrolimus/MMF	99.8%
- Steroids/tacrolimus/everolimus	0.2%

* Type of deceased donor, by brain death (DBD) or cardiac death (DCD), given as % of total donor kidneys, PRA: panel reactive antibodies, HLA: human leukocyte antigen, IQR: interquartile range defined as 25–75% percentile, SD: standard deviation. Steroids were tapered to stop 3–6 months after transplantation.

For graft loss other than death (10%), rejection constituted the major cause of graft loss but the frequency was only 5% (Table 2). In this population, the frequencies of re-transplantation (9%) and a positive PRA (7%) were relatively low and the median number of HLA mismatches was three.

### 3.2. Pre-Emptive Living Donor Kidney Transplantation Results in Superior Survival

Recipient survival after living donor transplantation was significantly but only marginally better compared to the deceased donor group (Figure 1). Dialysis before transplantation had a major impact on recipient survival with a median survival of 6.3 years in the non-pre-emptively transplanted group (n = 70) compared to 11.2 years in the pre-emptively transplanted group (n = 73, *p* < 0.001, Figure 2). A similar trend was observed in the recipients of a deceased donor kidney (*p* = 0.16, Figure 2) but the number of pre-emptively transplanted patients was relatively small in this subgroup (n = 30).

Multivariate regression analysis showed that only pre-emptive transplantation was highly significantly associated with survival after transplantation (HR for death for pre-transplant dialysis was 2.22 (1.47–3.35), *p* < 0.001, Table 3). The underlying causes of ESRD were similarly distributed in the living and deceased kidney donor groups (Appendix A), but the recipients of living kidney donors who were pre-emptively transplanted more frequently had a diagnosis of renovascular nephropathy and less frequently diabetic nephropathy. However, a diagnosis of diabetic nephropathy was not significantly associated with patient survival both in a univariate and multivariate model (Table 3).

The negative effect of dialysis before a kidney transplant on recipient survival (Figure 3) was found to be related to a substantially increased risk for infection-related death (HR 3.80, 95% CI 1.46–9.96), cardiovascular-related death (HR 3.35, 95% CI 1.16–9.71) but not death because of malignancy (*p*-value 0.4).

### 3.3. Graft Survival

Death-censored graft survival (DCGS) was superior for recipients of a living donor kidney who had a 5-year DCGS of 98% compared to 85% in deceased donor kidney recipients (Figure 1). This difference was driven by a higher graft loss within the first year after transplantation in the deceased donor kidney group (0% vs. 10%). In recipients of a deceased donor kidney, the causes of graft loss other than death within the first year after transplantation (n = 19) were predominantly rejection (9/19, 47%, n = 7 vascular T-cell-mediated rejection) and primary non-function (7/19, 35%). Vascular T cell mediated rejections were most frequently documented (Appendix A). The incidence of vascular rejection was 7% in the living donor group vs. 13% in the deceased donor group (*p* = 0.11). Of note, >90% of all rejection episodes occurred in the first 6 months after transplantation, confirming a previous observation that increasing age is associated with earlier plateauing of the cumulative incidence of rejection [7]. Dialysis before transplantation had a significantly negative impact on DCGS in recipients of a deceased donor kidney. When comparing the dialysis with the pre-emptive group, the difference in 1-year and 5-year graft survival was 90% vs. 97% and 84% vs. 92%, respectively. However, the patients pre-emptively transplanted with a deceased donor kidney significantly more often received a fully HLA-matched kidney (24% vs. 3%, *p* < 0.001) and from a younger donor (mean 53.5 vs. 65.0 years, *p* < 0.01) while cold ischemia time (mean 11.7 h for both groups) and type of deceased donor (brain death or not) did not differ. The superior graft survival for the living donor kidney recipients was unaffected by the pre-emptive status (5-year graft survival 98% non-pre-emptive vs. 97% pre-emptive group). Multivariate regression analysis showed that increasing age of the donor, deceased donor kidney, dialysis before transplantation and a positive PRA were all associated with decreased DCGS (Table 4). Within the first year, the effect of a positive PRA was most pronounced in the DD kidney group with a first-year DCGS of 97% in the PRA-negative group vs. 87% in the PRA-positive group (*p* < 0.001). Of note, graft loss other than death was predominantly seen for kidneys from donors older than 60 years (Figure 4), emphasizing donor age as an important determinant of graft survival in the deceased donor kidney group.

## 4. Discussion

The main findings of this study evaluating the results of kidney transplantation in patients of 70 years and older are a superior graft survival after transplantation with a living donor kidney and a major negative impact of dialysis before transplantation on recipient survival.

Concerning the latter finding, it has been shown in numerous publications that pre-emptive transplantation is associated with better graft survival and recipient survival [14,15,16,17]. However, the impact of pre-transplant dialysis on transplantation outcomes in the modern era of immune suppression in recipients of 70 years and older has not been documented in detail.

A recent publication showed data on patient and graft survival of 171 kidney transplant recipients of 70 years and older from a north-west French registry [18]. The vast majority (98%) received a deceased donor kidney and 85.8% of patients were on dialysis before transplantation. Their data show a death-censored graft loss of 17.4% in the first year after transplantation which is substantially higher than our 10%. The latter percentage is similar to the average percentage for this age group in Dutch transplantation centers [8]. This difference in first-year graft survival is likely explained by the higher mean donor age of 70 years (compared to 65 years in our study) and the fact that virtually all patients received expanded criteria donors in the French study. After one year, the Kaplan–Meier graft survival curves censored for death were relatively slowly decreasing, in line with our results. This indicates that, after the first year after transplantation, graft loss other than death is relatively rare in this elderly population, irrespective of the type of donor kidney.

It is known that the risk for acute rejection decreases progressively with age of the recipient while on the other hand kidneys from older donors are more prone to elicit a rejection. On average, the recipient’s age, that is, the immunological age, is a stronger predictor of acute rejection than the donor age [19]. In our study, the rate of acute rejection was 22% with a low overall rate of antibody-mediated rejection (either acute or chronic-active antibody-mediated rejection) of 4.6% over the whole follow-up period. How rejection episodes translate into graft loss is usually not recorded but based on this study it is clear that, in particular, recipients of an older (>60 years) deceased donor kidney were at risk for rejection-related graft loss. In this respect, the vascular type of rejection (T-cell-mediated rejection type 2–3) was a dominant factor which is known to be associated with HLA matching and previous HLA sensitization (as, e.g., shown by a positive PRA of >5%) [20]. In our protocol we do not use T cell depletion agents such as alemtuzumab or ATG which may decrease the incidence of vascular rejection but at the unwanted cost of more serious infections in this in general frail patient group [21,22].

Analysis of the cause of graft failure uncensored for death revealed death as the predominant event with relatively few graft losses because of, e.g., rejection. This outcome is in line with a previous publication [7] showing that elderly recipients have a substantial decreased risk for rejection-related graft loss compared to younger recipients, while death is a major competitive risk factor. Of note, about half of the total number of early failures in deceased donor kidney transplantation could be attributed to acute rejection which seemed to be less in pre-emptively deceased donor transplanted patients. This is most likely because of the Eurotransplant allocation algorithm: allografts below 65 years of age are prioritized for recipients of any age group when recipients have nil HLA mismatches on A, B and DR. As a consequence, elderly candidates can virtually only receive a transplant pre-emptively outside the ESP and with HLA-matched younger donors.

Similar to our data, the French group reported infection as a major cause of death, particularly within the first year of transplantation. Our protocol with steroid withdrawal at 3 months and non-lymphocyte-depleting induction therapy appears justified given this high infection-related death rate and the low incidence of rejection. The optimal immune-suppressive regimen in the elderly patients, in particular reduced maintenance immune suppression, is a matter of debate with a clear knowledge gap [23]. Recently, it was shown in a small RCT that immunologically low-risk elderly recipients could be safely managed with tacrolimus monotherapy one year after transplantation. The results showed significantly fewer infections and a much better COVID vaccination response while medication adherence improved with fewer gastro-intestinal side effects [24]. Though these results indicate that the aged immune system of the elderly allows for and benefits from less intense immune suppression, confirmation in a larger study is required.

Of interest is the effect of dialysis before transplantation on the causes of death post-transplantation, showing a specific increase in both cardiovascular disease and infection-related death in the non-pre-emptively transplanted compared to pre-emptively transplanted patients. This underlines the detrimental effect of dialysis and/or loss of renal function on vascular health and the known effects of progressive eGFR loss on premature aging of the immune system, which is even more prominent in dialysis patients [25,26].

The median survival of pre-emptively transplanted recipients of a living donor kidney was surprisingly good and the negative effect of dialysis before transplantation is much larger than in most studies. To put data into perspective, in 2015 the average survival of the general Dutch population aged 70 was 15 years and aged 75 this was 11 years (Central Bureau of Statistics: www.cbs.nl (accessed on 30 November 2023)). The average survival was 11.6 years in the pre-emptively transplanted patients (median age 73 years) which is remarkably good in these recipients with a clinical history of hypertension and/or diabetes mellitus. Some of the observed survival advantage of pre-emptive transplant recipients may be due to lead time bias. Lead time bias may cause a perceived survival advantage in pre-emptively transplanted patients, as their post-transplant survival time is calculated from an earlier starting point than non-pre-emptively transplanted recipients. The effect of lead time bias was assessed in a study by Irish et al. in which a pre-emptively transplanted group of patients was compared to patients on dialysis for less than 6 months and no difference in survival was found [27]. However, the median age of the recipients was 43 years and patients receiving >6 months of dialysis before transplantation were not included. Therefore, these results are not readily applicable to the older recipients included in this study. In a recent study, an intention-to-treat analysis was chosen to account for lead time bias in a group of kidney transplant recipients of 70 years or older (UNOS database 2014–2021) [4]. This analysis showed that patients who were transplanted pre-emptively, whether with a deceased donor or a living donor kidney, had a significantly better survival than those who were transplanted after initiating dialysis.

Given the excellent results of living donor kidney transplantation in the elderly, our transplantation center emphasizes timely referral in order to have enough time for living donor assessment and pre-emptive transplantation. This results in a relatively high number of LD kidney transplantations compared to other centers, although LD transplantation rates could be further optimized. In a recent publication, about 30% of nephrologists considered age as an important reason for not referring to a transplantation center [28], while the KDIGO guideline clearly states that age by itself is not a barrier [29]. Late referral is one of the reasons why a substantial part of elderly patients (30% in a previous study from our center) are delisted before transplantation [30,31]. Also, old age is associated with lower utilization of living donor kidney transplantation [30]. Agism and socio-economic factors seem important determinants for the underused option of kidney transplantation in the elderly [14,28,31,32].

A limitation of the current study is that we cannot account for bias in recipient selection which may result in different outcomes among centers. However, although our center does have a pro-active policy for kidney transplantation in the elderly, the pre-transplantation assessment of candidates is carried out in accordance with the KDOQI guideline.

## 5. Conclusions

This large cohort of elderly kidney transplant recipients shows excellent long-term death-censored graft survival and superior patient survival in those receiving a living donor kidney transplant without previous dialysis.

## Figures and Tables

**Figure 1 jcm-13-01853-f001:**
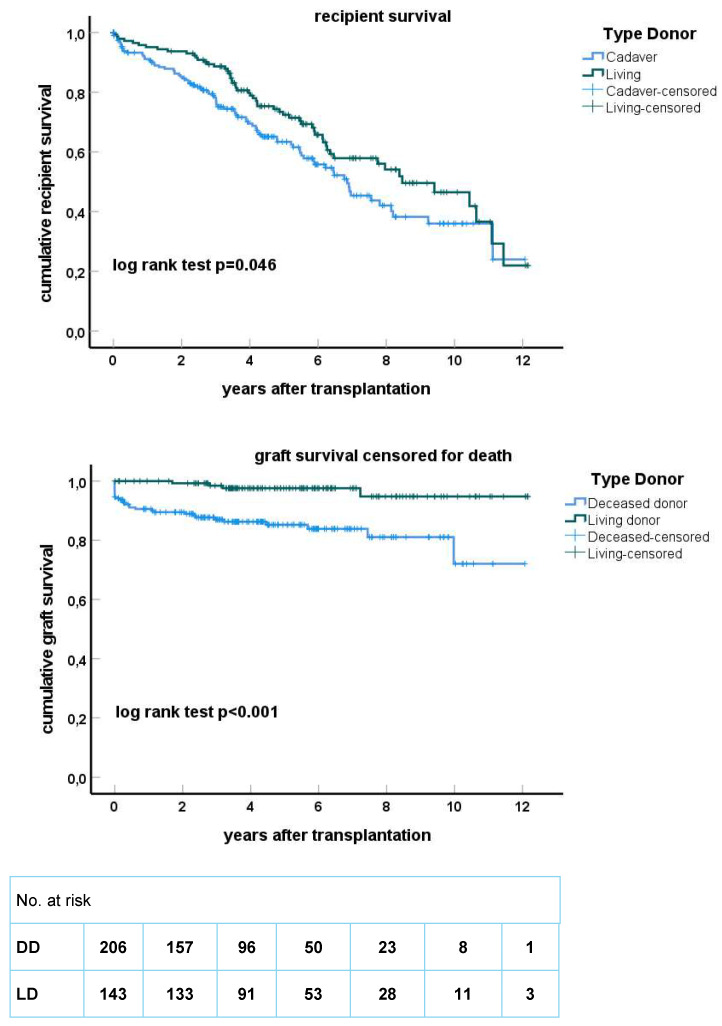
Kaplan–Meier curves for recipient survival (**upper figure**) and death-censored graft survival (**lower figure**) stratified for living donor (LD) and deceased donor (DD) kidney grafts. Recipients at risk at every 2 years after transplantation are given below the X-axis.

**Figure 2 jcm-13-01853-f002:**
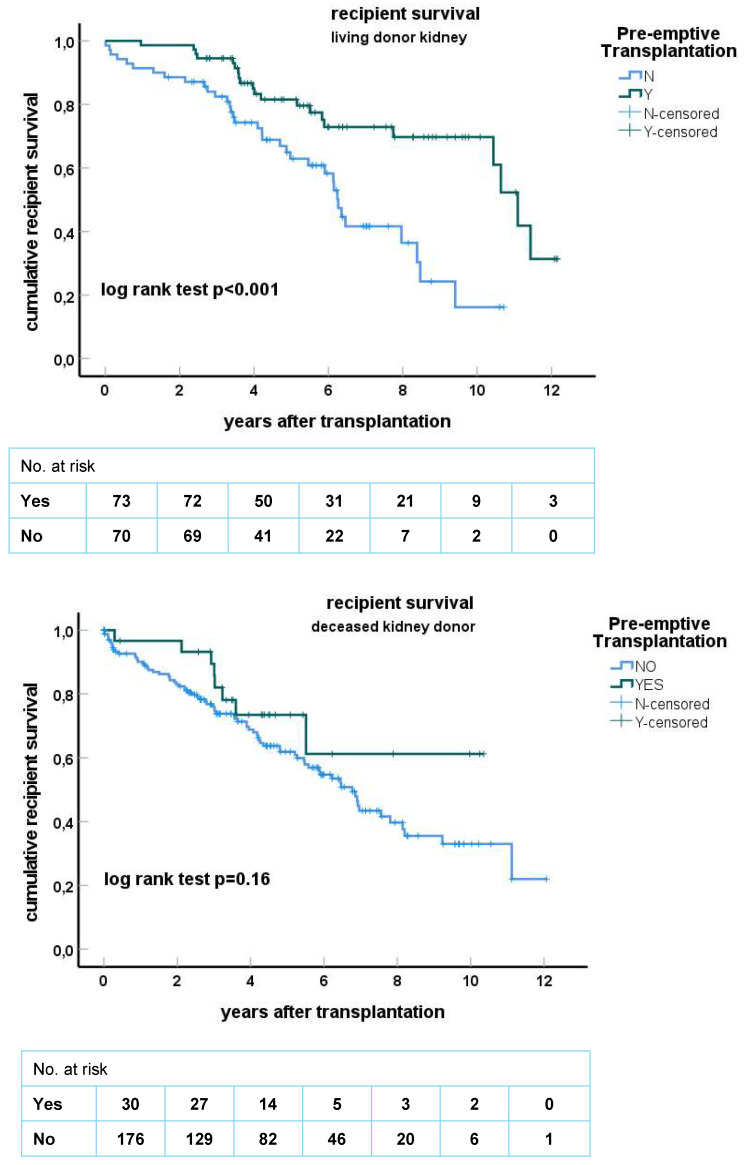
Kaplan–Meier curves for recipient survival for living donor (LD, **upper figure**) and deceased donor (DD, **lower figure**) kidney grafts stratified for pre-emptive transplantation or not. Recipients at risk at every 2 years after transplantation are given below the X-axis. The *p*-value for log-rank testing for difference between the curves is shown.

**Figure 3 jcm-13-01853-f003:**
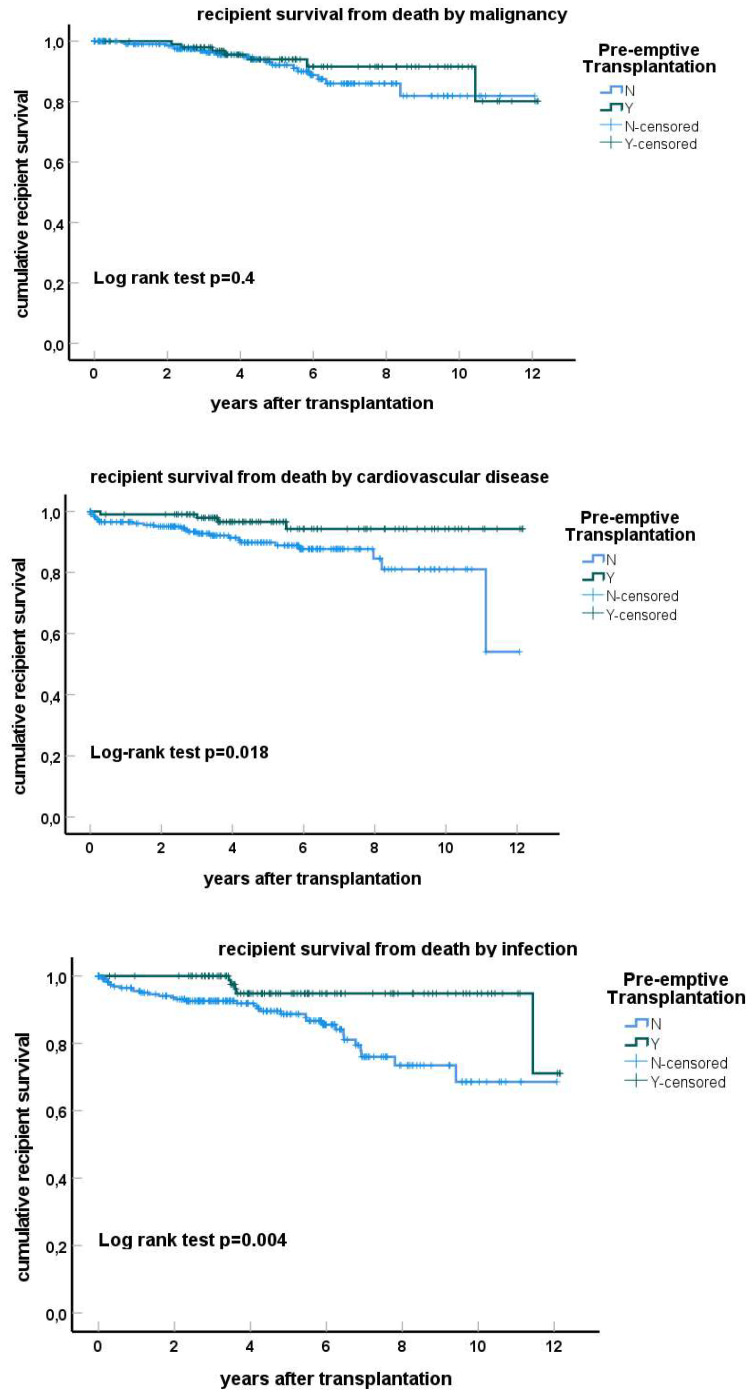
Kaplan–Meier curves for recipient survival from death by malignancy, cardiovascular disease and infection stratified for pre-emptive transplantation or not. The *p*-value for log-rank testing for difference between the curves is shown.

**Figure 4 jcm-13-01853-f004:**
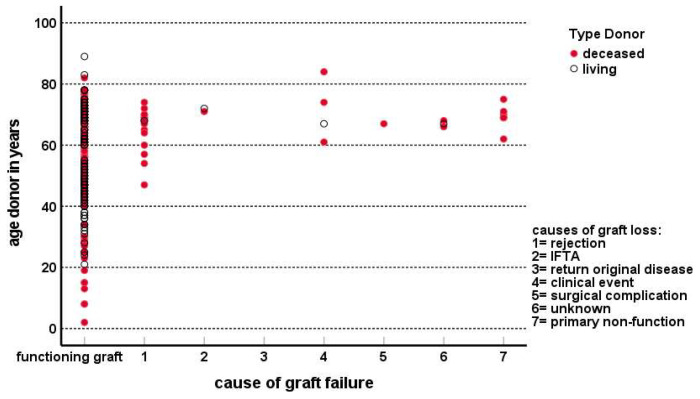
The cases of graft loss other than death are shown in relation to the age of the donor kidney and the type of donor kidney (living donor or deceased donor). The category 0 on the x-axis contains all cases with a functioning graft at follow-up or graft loss because of death.

**Table 2 jcm-13-01853-t002:** Causes of graft failure in 349 kidney transplant recipients of 70 years or older during follow-up of 7 to 13 years.

Follow-up in years, median (interquartile range) Death with functioning graft Number of graft losses other than by death	4.3 (2.6–6.4) 140 (40.1%) 33 (9.5%)
Graft loss by:	
- rejection; total number - T-cell-mediated rejection - antibody-mediated rejection - interstitial fibrosis/tubular atrophy - recurrence of original disease - kidney injury/disease - peri-operative complications - unknown - primary non-function	16 (4.6%) 11 (3.1%) 5 (1.4%) 2 (0.5%) 0 (0%) 4 (1.1%) 1 (0.5%) 3 (0.8%) 7 (2.0%)

**Table 3 jcm-13-01853-t003:** Univariate and multivariate Cox regression analysis for outcome death after kidney transplant.

	Univariate Analysis		Multivariate Analysis	
	HR (95% CI) *	*p*-Value	HR (95% CI)	*p*-Value
Age recipient per year	1.07 (1.01–1.13)	0.013	1.06 (1.00–1.12)	0.05
Age donor per year	1.00 (0.99–1.01)	0.88	-	-
Type of donor kidney DD vs. LD **	1.4 (1.01–2.00)	0.042	-	-
Pre-emptive transplantation no vs. yes	2.22 (1.47–3.35)	<0.001	2.22 (1.47–3.35)	<0.001
Diabetes mellitus yes vs. no	1.43 (0.97–2.09)	0.068	-	-

* HR (95% CI): hazard ratio with 95% confidence interval, ** DD: deceased donor, LD: living donor.

**Table 4 jcm-13-01853-t004:** Univariate and multivariate Cox regression analysis for outcome graft loss censored for death after kidney transplant.

	Univariate Analysis		Multivariate Analysis	
	HR (95% CI) *	*p*-Value	HR (95% CI)	*p*-Value
Age recipient per year	0.97 (0.85–1.10)	0.68	-	-
Age donor per year	1.05(1.02–1.09)	0.004	1.05 (1.01–1.08)	0.01
Type of donor kidney DD vs. LD **	6.21 (2.19–17.66)	<0.001	5.30 (1.85–15.1)	0.002
Pre-emptive transplantation no vs. yes	2.43 (0.99–5.89)	0.051	2.22 (1.47–3.35)	<0.001
Previous transplantation yes vs. no	1.12 (0.49–2.54)	0.78	-	-
PRA *** high vs. low	2.53 (1.13–5.66)	0.023	3.01 (1.38–6.95)	0.006
Total HLA mismatches	1.12 (0.89–1.40)	0.31	-	-

* HR (95% CI): hazard ratio with 95% confidence interval, ** DD: deceased donor, LD: living donor, *** PRA: panel reactive antibodies (high is defined by PRA > 5%), HLA: human leukocyte antigen

## Data Availability

The original contributions presented in the study are included in the article/Appendix A, further inquiries can be directed to the corresponding author.

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
