# Peer review of "Transplant Candidates of 70+ Years Have Superior Survival If Receiving Pre-Emptively a Living Donor Kidney"

_jcm, 2024, doi:10.3390/jcm13071853_

Round 1

Reviewer 1 Report

Comments and Suggestions for Authors

I enjoyed reading this article and learned a great deal.

Comments on the Quality of English Language

In English (American), transplantation refers to the overall process or a non-countable process whereas transplant refers to a countable process or a procedure. In the paper, the word transplantation is used far too often.

Additionally, immunosuppression is the term used not "immune suppression".

In the abstract, the word "transplantation" works where it was used.

Line 22: There is an "in" that needs to be removed.

Line 30: Please replace "transplantation" with "transplant" and "immune suppression" with "immunosuppression".

Line 31: Please replace "transplantations" with "transplants".

Line 42: Please replace the comma after [5] with "and".

Line 43: Please replace "transplantation" with "transplant".

Line 45: Please replace "immune suppression" with "immunosuppression".

Line 63: Please replace "transplantations" with "transplants".

Line 68: Please replace "Transplantations" with "Transplants".

Line 84: I do not understand the phrase "in case of" and would prefer "due to".

Line 95: IFTA stands for "interstitial fibrosis and tubular atrophy". 

Lines 96-100: I am confused about these two sentences. First, the term "for cause kidney biopsy" is to differentiate, I assume, that type of biopsy from "protocol biopsies". Since the authors do not do protocol biopsies, I would rather they state that the only kidney biopsies done are for making a diagnosis when the graft is failing or there is proteinuria. The other problem with the two sentences is I do not understand why the sentence from Line 96-98 is needed as the sentence in Lines 98-100, seems to include all 33 of the possible causes of graft loss.

Line 100: "Table 1" should be "Table 2".

In Table 2, "tubulus" should be "tubular".

Is it the case that no kidney biopsy was done in the 3 patients having an "unknown" cause of kidney failure?

Were the kidneys biopsied in the 7 patients with primary non-function?

Line119: There is a "13" after "As expected". is that supposed to be there?

For the patients who were on dialysis at the time of their transplant, were they all on hemodialysis or were some on peritoneal dialysis. Did those on peritoneal dialysis do better than those on hemodialysis prior to their  kidney transplant?

In Figure 2, I feel all the times Transplantation is used, Transplant would be better.

Line 155: Please replace "ESKD" with "ESRD".

Line 156: The underlying causes of ESRD were reported to be in Table S2, but that is not the case. That data is in Table 1, I believe. In Table S2 are the Banff rejection classifications. (Please add a space between "Rejection" and "by" in the title, by the way.)

Line 157: Please rewrite the sentence to say "the recipients of living donor kidneys who" from "the living kidney donors who".

Lines 211-212: Please replace "transplantation" with "transplant".

Line 265: "Statitistics" should be "Statistics".

Tabe 3: Please replace "transplantation" with "transplant" after preemptive.

Line 164: Please replace "transplantation" with "transplant".

Figure 3: Please replace "transplantation" with "transplant" in 7 places on the figures and in the legend.

Figure 1: Please replace "transplantation" with "transplant" in 3 places on the figures and in the legend.

Reviewer 2 Report

Comments and Suggestions for Authors

This is an interesting large single center study describing the results of renal transplantation in elderly patients.  This reviewer wishes to suggest several modifications which may provide helpful information for readers and researchers.

 As I review the manuscript, the recurring concern is potential underlying differences in the studied population, which potentially bias the reported results.   I wonder if the top of Table 1 should be modified to show clearly the if baseline similarities and differences exist between the 4 distinct patient categories reported: LRD preemptive, LRD post dialysis, DCD pre-emptive, DCD post dialysis.  Data presented as IQR’s maybe more instructive than medians and range.

A few other questions come to mind:

1/ Does dialysis duration have an effect on patient or graft survival post-transplant.

2/ I appreciate the comparison in the discussion of life span in portions of the report cohort compared to the age corrected general population.  Do you have sufficient data to compare patient survival for the post- dialysis transplanted group compared to similarly aged patients remaining on dialysis?

3/ There are several mentions of expanded criteria donors.  Were many of the primary non-functions and/or rejections from this group?  I assume that few, if any, of the LRD’s were expanded criteria donors.

4/  Although it is not part of the message of this manuscript (and not necessary), I wonder if in a population this large, the mortality risk of the transplant itself can be identified?

5/ Table S2 shows 77 rejection episodes and the text suggests only 4.2% of surviving patients had rejections.  Thus, was rejection a leading cause of mortality and/or graft loss?

6/ I assume that the only reason very young DCD grafts were provided to older recipients was “full-house” HLA matching.  If so, these recipients had a double dose of good luck and would be expected to do relatively very well.

Less important points:

1/  In several of the graphs  I assume the hash marks indicate when mortality events occur—but these are very hard to see clearly in the pdf provided.

2/ Table S2 shows Banff rejection data rather than primary diagnosis as indicated in line 156.

3/ Multiple incomplete references (e.g. #3, 7, 13, 20, etc)

Author Response

This is an interesting large single center study describing the results of renal transplantation in elderly patients.  This reviewer wishes to suggest several modifications which may provide helpful information for readers and researchers.

We wish to thank the reviewer for his appreciation of the manuscript and the careful review.

 As I review the manuscript, the recurring concern is potential underlying differences in the studied population, which potentially bias the reported results.   I wonder if the top of Table 1 should be modified to show clearly the if baseline similarities and differences exist between the 4 distinct patient categories reported: LRD preemptive, LRD post dialysis, DCD pre-emptive, DCD post dialysis.  Data presented as IQR’s maybe more instructive than medians and range.

We agree that medians with IQR’s may be more instructive and have changed this.

We deliberately did not  make a Table with 4 subgroups as we do not think this will contribute to the readability of the manuscript. The DCD pre-emptive group is also relatively small and as stated in the manuscript has received in general very well matched kidneys because of the Eurotransplant policy. More fundamentally, to consider the whole group of elderly recipients and perform a multivariate analysis is in our opinion the best approach to show the relation with graft and recipient survival. Having said that, there may be differences between populations that we cannot account for. In some studies it was noted that pre-emptively transplanted recipients had a higher socio-economic status and this may translate in a healthier lifestyle. Also, the referring nephrologist probably made a selection in recipients that may or may not be fully in line with current guidelines. The key message we want to convey with this study is that the results of pre-emptive kidney transplantation in the 70+ transplant candidates are surprisingly good in terms of death-censored graft loss and patient survival is superior if pre-emptively transplanted. Probably this is largely due to residual renal function and no “vascular stress” of the dialysis treatment but there may be many confounders. However, the bottom line is that kidney transplantation in the 70+ group may yield (very) good outcomes and should be performed more frequently.

A few other questions come to mind:

1/ Does dialysis duration have an effect on patient or graft survival post-transplant.

Dialysis duration has no effect on graft survival but is known to decrease patient survival. Data from the Dutch Organ Transplant Registry (NOTR) show that the effect of dialysis duration is present (but modest) from year 1 onwards as compared to no or <1 years of dialysis. In our cohort, the median duration of dialysis is relatively short with a median of 1.8 years and (most likely for that reason) we could not find a significant association with patient survival.

2/ I appreciate the comparison in the discussion of life span in portions of the report cohort compared to the age corrected general population.  Do you have sufficient data to compare patient survival for the post- dialysis transplanted group compared to similarly aged patients remaining on dialysis?

Unfortunately, we do not have access to such data and we cannot make this comparison. Such a comparison also requires a different statistic approach taking care for phenomena such as “immortal time lead bias”. In the cited paper by Sibulesky et al, this was studied for the 70+ recipients by an intention-to-treat analysis showing that candidate survival was significantly improved for those transplanted preemptively versus being on dialysis (hazard ratio 0.59; confidence interval, 0.56-0.63). We added some lines in the manuscript discussion to highlight this result.

3/ There are several mentions of expanded criteria donors.  Were many of the primary non-functions and/or rejections from this group?  I assume that few, if any, of the LRD’s were expanded criteria donors.

None of the LRD’s were expanded criteria donors. As can be seen from figure 4, the primary non-functions and graft losses because of rejection were in the 60+ kidney donor group. Age of the donor is the most important factor associated with PNF and graft loss.

4/  Although it is not part of the message of this manuscript (and not necessary), I wonder if in a population this large, the mortality risk of the transplant itself can be identified?

Interesting question but difficult to perform such an analysis. The survival curves do not convincingly show a short term excess mortality and almost run a straight line. Of course this is not very strong evidence but the survival dip shortly after transplantation in the older (USA) literature is not observed in our 70+ transplant candidates.

5/ Table S2 shows 77 rejection episodes and the text suggests only 4.2% of surviving patients had rejections.  Thus, was rejection a leading cause of mortality and/or graft loss?

In table 2 it is shown that 173 recipients lost their graft; 140 because of death with functioning graft and 33 graft loss for other reasons. Sixteen of these 33 graft losses were because of rejection (4.6% of total number of 349 patients, compared to 40.1%  patients who died with a functioning graft). Rejection is no leading cause for mortality but does have a relatively large contribution (but not in absolute numbers) to graft loss censored for death.

6/ I assume that the only reason very young DCD grafts were provided to older recipients was “full-house” HLA matching.  If so, these recipients had a double dose of good luck and would be expected to do relatively very well.

This assumption is correct and their graft survival censored for death is excellent (see also figure 4 with no graft loss in the young DD donor group).

Less important points:

1/  In several of the graphs  I assume the hash marks indicate when mortality events occur—but these are very hard to see clearly in the pdf provided.

This assumption is correct. The figures will be improved upon acceptance of the manuscript.

2/ Table S2 shows Banff rejection data rather than primary diagnosis as indicated in line 156.

The underlying causes of ESRD of the recipients are shown in Table S1. This has now been corrected.

3/ Multiple incomplete references (e.g. #3, 7, 13, 20, etc)

 Where needed this has been corrected.